# Ethanol Biofuel Cells: Hybrid Catalytic Cascades as a Tool for Biosensor Devices

**DOI:** 10.3390/bios11020041

**Published:** 2021-02-04

**Authors:** Jefferson Honorio Franco, Shelley D. Minteer, Adalgisa R. De Andrade

**Affiliations:** 1Department of Chemistry, Faculty of Philosophy Sciences and Letters at Ribeirão Preto, University of São Paulo, Ribeirão Preto, Sao Paulo 14040-901, Brazil; jeffersonhfranco@gmail.com; 2Department of Chemistry, University of Utah 315 S 1400 E Rm 2020, Salt Lake City, UT 84112, USA

**Keywords:** biofuel cell, hybrid system, biosensor

## Abstract

Biofuel cells use chemical reactions and biological catalysts (enzymes or microorganisms) to produce electrical energy, providing clean and renewable energy. Enzymatic biofuel cells (EBFCs) have promising characteristics and potential applications as an alternative energy source for low-power electronic devices. Over the last decade, researchers have focused on enhancing the electrocatalytic activity of biosystems and on increasing energy generation and electronic conductivity. Self-powered biosensors can use EBFCs while eliminating the need for an external power source. This review details improvements in EBFC and catalyst arrangements that will help to achieve complete substrate oxidation and to increase the number of collected electrons. It also describes how analytical techniques can be employed to follow the intermediates between the enzymes within the enzymatic cascade. We aim to demonstrate how a high-performance self-powered sensor design based on EBFCs developed for ethanol detection can be adapted and implemented in power devices for biosensing applications.

## 1. Enzymatic Biofuel Cells (EBFCs)

In 1911, Potter et al. demonstrated the first biofuel cell (BFC), which contained yeast cells at the anode and oxidized glucose [1]. Decades later, in the 1960s, Kimble and collaborators showed the initial proof-of-concept of “enzymatic biofuel cells” (EBFCs) by fixing glucose oxidase (GOx) on the surface of an electrode. The fact that an enzyme was able to produce electricity was essential to raising interest in the field of bioelectrochemistry [2].

BFCs resemble conventional fuel cells. The main difference is that metallic catalysts are replaced with enzymes and/or microorganisms in EBFCs and microbial fuel cells, respectively [3]. In BFCs, the biocatalyst at the anode promotes oxidation of the target fuel. The preferred reaction at the cathode is oxygen reduction, which can also be catalyzed by enzymes, such as laccase (Lac) or bilirubin oxidase (BOX) [4,5], or by microorganisms [6]. To produce electricity, electrons released at the anode reach the cathode via an external circuit, as illustrated in Figure 1. The useful energy generated in the system depends on the difference between the anode and the cathode potentials [7].

Advances in the use of isolated enzymes as biocatalysts have been made thanks to (i) the various reactions that enzymes can catalyze with unequaled specificity even in the presence of impurities [8], (ii) the greater potential of isolated enzymes for in vivo applications [9], (iii) the higher power density values provided by EBFCs as compared to microbial fuel cells (where the microorganism cell membrane limits mass and electron transfer) [10], and (iv) the possibility of operating EBFCs at neutral pH. However, the experimental power and energy density of these systems are lower than the theoretical performance [3,11,12]. Thus, to optimize self-energy generation, the factors that contribute to decreased cell potential must be optimized [7]:E_cell_ = (Δ_Ec-Ea_) − Δη − ΣΩ(1)
where Ec and Ea are the thermodynamic potentials for the cathode and the anode, respectively; Δη is the overvoltage of the anodic and cathodic reactions (Δη takes the slow kinetics and the mass transfer rate into account); and ΣΩ is the sum of the internal resistances of the cell [7]. Energy harvesting and the power output of EBFCs must be optimized so that this technology can become as useful as commercial batteries and conventional fuel cells. Therefore, current efforts in the field of BFCs have been focused on developing new methodologies and materials integrated with enzymes to increase the useful life, power density output, and consumption of all the energy potentials of the biofuel until it is completely oxidized to CO_2_ [13,14].

## 2. Enzymatic Cascades

In recent years, research into EBFCs has focused on understanding how enzymes function on electrode surfaces and on developing methods for the application of enzymes on these surfaces. To enhance the power density of EBFCs, the degree of fuel oxidation must be increased so that all electrons can be harvested from complex fuels [15]. EBFCs generally employ enzymes to collect energy from biofuels [3,11,16]. These enzymes catalyze the oxidation of various fuels at room temperature and in mild aqueous environments, offering high theoretical efficiency while producing nontoxic reaction residues [12,15,17,18].

To increase the degree of fuel oxidation, multiple enzymes have been immobilized on electrode surfaces [19]. These systems can oxidize more complex fuels, thereby enhancing power density and energy generation [12,20,21]. Nevertheless, mechanically and chemically stable layers that employ a large number of enzymes almost always afford poorly stable bioelectrodes [12]. Different specific operating conditions (pH, temperature, substrate specificity, and electrolytes) of the enzymes in the enzymatic cascade limit film stability. Our group has shown that, compared to a bi-enzymatic anode, a multi-enzymatic system involving six enzymes did not furnish higher power density for ethanol bioelectrooxidation [22] because immobilization of the six enzymes on the same matrix limited their functions and culminated in lower bioelectrocatalytic rate [22].

Therefore, for the theoretical energy (all the electrons) of a fuel to be completely harvested, new methodologies that rely on new materials (e.g., nanostructures, hybrid catalysts, and modified catalysts) integrated with enzymes must be developed [23].

## 3. New Trends in EBFCs: Hybrid Cascades for Ethanol Electrooxidation Pathways

Many research groups, including ours, have been investigating EBFCs [21,24,25,26,27]. Using ethanol as fuel is attractive because it is low in toxicity, has a high energy density (8.6 kWh kg^−1^), and is produced worldwide [24,26,27,28]. This renewable fuel has promising characteristics for the development of energy conversion devices, especially for application in biosensors [10,29,30,31]. However, ethanol oxidation through a multi-enzymatic cascade is complicated: a large number of enzymes are required for this fuel to be completely oxidized [12,22]. In an attempt to obtain improved results during the development of BFCs, we have recently investigated electrochemical ethanol oxidation to CO_2_ by using a hybrid biocatalyst consisting of less specific organic catalysts and enzymes integrated into nanostructured materials [32,33,34]. The results regarding the collection of the maximum number of electrons from different alcohols such as glycerol [35,36] and ethanol [32,33] were interesting. 2,2,6,6-Tetramethylpiperidine-N-oxyl (TEMPO), which oxidizes alcohols and aldehydes, was employed as an organic catalyst [29,30]. Electrochemical oxidation with TEMPO generates a hydroxylamine, which produces the nitroxyl radical and completes the catalytic cycle [37,38]. Nevertheless, TEMPO does not remove all the electrons from the substrate because it cannot cleave carbon–carbon bonds. In this situation, complete alcohol oxidation could be boosted by employing hybrid systems that combine TEMPO with enzymes that can specifically cleave C–C bonds [32,36,37,38]. In fact, nanostructured materials are being increasingly studied to enhance the power density output of EBFCs. The modern design of catalysts for BFCs must incorporate nanoscale materials (carbon nanotubes, nanofibers, graphene, and nanocomposites) into the bioelectrode structure [39]. Research into nanostructured materials has demonstrated excellent entrapment of the immobilized macromolecule: these materials provide a more adequate environment for enzymatic anchoring, allowing for higher enzyme loading and thus increasing the kinetic process efficiency. The presence of these nanomaterials improves the electrical contact between the active sites of the enzymes and the electrode surface, enhancing the bioelectrode electroactivity [40].

Within the field of BFCs, multi-walled carbon nanotubes (MWCNTs) are attractive due to their outstanding features: they display (i) large specific surface area, (ii) high mechanical strength, (iii) high conductivity, (iv) good biocompatibility, and (v) antifouling properties [41]. Moreover, MWCNTs can be electrically connected with many redox enzymes [42] or can be modified with functional groups for further immobilization of biomolecules [43,44].

Given that the combination of nanoparticles with biomaterials provides systems with improved electronic and catalytic properties, hybrid nanobiomaterials have been developed to achieve higher catalytic power and power densities [45].

## 4. Complete Ethanol Oxidation by Systems Based on Hybrid Enzymatic Electrodes and Organic Catalysts

We were the first to report on a hybrid system containing TEMPO and enzymes for use in an EBFC [32]. This bi-catalytic system combined carboxylated multi-walled carbon nanotubes (MWCNT-COOH), TEMPO-modified linear poly(ethylenimine), and alcohol (ADH) and aldehyde (AldDH) dehydrogenase immobilized on an linear poly(ethylenimine) (LPEI) backbone crosslinked carbon electrode. This system cleaved the acetic acid C–C bond to give CO_2_ as the final product (12 electrons). In other words, this immobilized hybrid bi-catalytic system yielded high electrochemical oxidation rates and complete ethanol oxidation to CO_2_. The long-term (12 h) electrolysis chromatographic data revealed high CO_2_ yields and confirmed total ethanol oxidation to CO_2_. The hybrid biofilm consisting of MWCNT/TEMPO-LPEI/ADH + AldDH improved the immobilization system and may be employed to build hybrid architectures for biosensors [32].

Seeking to improve catalytic systems further, our group later reported complete ethanol oxidation in hybrid systems containing the organic catalyst TEMPO and enzymes that can cleave the C–C bond, such as oxalate oxidase (OxOx) [33] and oxalate decarboxylase (OxDc) [34,46]. It was the first time that a hybrid system containing an organic catalyst and an oxidase/decarboxylase enzyme was reported to oxidize ethanol more efficiently than previously reported hybrid systems [34,46].

The hybrid system containing pyrene-TEMPO immobilized on the surface of MWCNT-COOH and OxDc deposited on a carbon cloth electrode deserves highlighting. This new bi-catalytic architecture provided EBFCs with a longer useful lifetime and higher power density values and electrocatalytic performance [33,34,46]. Despite the challenges in cleaving the C–C bond of the acetate intermediate, this new hybrid bifunctional enzymatic/organic electrocatalyst electron-harvesting design allowed 12 electrons to be collected per ethanol molecule and produced more energy through complete ethanol oxidation, as shown in Figure 2.

Scheme 1 depicts the proposed electrocatalytic cascade for ethanol oxidation when a hybrid bi-catalytic architecture is used [34]. The enzymatic pathway acts on the product formed by the TEMPO catalyst. The combination of pyrene-TEMPO and OxDc allows for high-energy production. Furthermore, the introduction of MWCNT-COOH improves the electron transfer rate between the enzymes and the electrode surface.

## 5. Analytical Techniques Employed with Ethanol EBFCs

Analytical techniques are critical tools for identifying and quantifying electrolysis products and for confirming important parameters such as mass transfer efficiency and kinetic rates at the electrode and in electron harvesting from the fuel. Although several groups have reported complete alcohol oxidation on the basis of electrochemical results [12,19,47], a reliable and sensitive technique is necessary to determine and quantify the products generated during fuel oxidation in BFCs. In the area of bioelectrochemistry, identifying such products is crucial because it paves the way for improving the construction of efficient devices.

Analytical techniques provide insight into how BFCs operate. We used high-performance liquid chromatography (HPLC) coupled with a refractive index detector (RID) to investigate ethanol BFC [21]. To achieve the best results for ethanol oxidation, we employed an Aminex HPX-87H (Bio-Rad) column in the isocratic mode and sulfuric acid (5 mmol L^−1^) at a flow rate of 0.6 mL min^−1^ as the mobile phase. We selected the Aminex HPX-87H column because it can identify and detect volatile fatty acids, carboxylic acids, and alcohols efficiently [48]. RID was chosen because it is an attractive and universal type of detector that can be coupled with HPLC. The resulting HPLC–RID system represents a promising, efficient, sensitive, and reliable analytical technique to quantify alcohol, carbohydrates, and carboxylic acids [48]. The RID offers numerous advantages, including stability, robustness, and versatility [49]. In this case, using a UV-detector only would not have been feasible because it cannot detect ethanol or some of its oxidation products. Product detection could be improved by employing a dual detector system in series ((UV-VIS) + RID) to obtain maximum information about the 11 possible products and substrate consumption. The first published investigation into an ethanol BFC employing HPLC results confirmed the electrochemical data and revealed that acetic acid (four electrons) was the only by-product [21].

When it comes to proposing a BFC device, the main challenge is the collection of all electrons from the fuel and its complete oxidation to obtain maximum energy during BFC operation. Although excellent results have been published, an analysis of the electrochemical results does not suffice to confirm complete fuel oxidation. Therefore, HPLC can be an essential tool to detect the CO_2_ formed in solution to confirm complete organic substrate oxidation. Detecting CO_2_ by an analytical technique is vital in the field of EBFCs because it demonstrates without doubt that the fuel has been completely oxidized. We have recently reported the detection of CO_2_ generation by HPLC [32]. To this end, we added 0.1 mol L^−1^ NaOH to the electrochemical cell after electrolysis, so that the RID detector could easily detect the resulting sodium carbonate as a negative chromatographic peak [32].

We also employed chromatographic results to confirm CO_2_ formation after long-term ethanol electrolysis at a hybrid electrode combining an organic catalyst, TEMPO, and the OxOx enzyme [33]. The products formed after electrolysis for 12 h confirmed that the hybrid electrode system (MWCNT-COOH/TEMPO-LPEI/OxOx) catalyzed multiple ethanol electrooxidation steps (Figure 3). Figure 3A,B display the results we obtained for the MWCNT-COOH/LPEI/bovine serum albumin (BSA) electrode system at 0 and 12 h, respectively. In the absence of TEMPO or an enzyme, no products emerged for the control electrodes. The results confirmed that MWCNT-COOH only acted to enhance the electron transfer and electrical contact between the active sites of the enzymes and the electron collector and that it was active in the oxidation pathway. The system containing only OxOx (MWCNT-COOH/LPEI/OxOx) in the presence of ethanol gave a similar result (Figure 3C). The enzymatic system afforded no product because this enzyme was not active for alcohol. Nevertheless, HPLC analysis showed that the OxOx enzyme cleaved the acetic acid C–C bond and yielded formic acid as a by-product (Figure 3D).

As expected [38,50,51], ethanol oxidation at the TEMPO-LPEI electrocatalyst (MWCNT-COOH/TEMPO-LPEI/BSA) produced acetic acid only. We have reported that TEMPO catalyzes ethanol oxidation, harvesting four electrons (4 e^−^) from this fuel (Figure 3E) [33]. To increase the number of harvested electrons, a hybrid system must be prepared by introducing an enzyme that can cleave the C–C bond (Figure 3F). Thus, the CO_2_ detected by HPLC (peak 4 in Figure 3) confirmed that ethanol C–C bond cleavage and collection of the 12 electrons were possible (Figure 3).

Analytical techniques such as nuclear magnetic resonance (NMR) and gas chromatography (GC) coupled with a thermal conductivity detector (TCD) have been proposed to detect electrolysis intermediates and products during the study of the catabolic steps of fuel oxidation [34,36,37].

Even though the HPLC–RID technique provides good quantitative results, the difficulty in detecting volatile compounds such as acetaldehyde during the first catabolic step of ethanol oxidation has been the main reason for seeking new analytical techniques. Another issue is the need for obtaining high-resolution peaks to avoid misdetection of the target analytes. In this context, NMR is a powerful analytical technique to determine structural properties and to quantify and identify various compounds without the drawbacks of decomposition, sample modification, oxidation during analysis, or even total matrix loss, for instance [52,53]. In addition, compared to other analytical detection methods, such as ultraviolet (UV), infrared (IR), and refraction index, NMR can discriminate between compounds of similar structures by means of ^1^H NMR or ^13^C NMR analyses [54]. Our group identified CO_2_ by NMR after complete glycerol oxidation [35,37] and after long-term ethanol electrolysis in an ethanol BFC system (amino-TEMPO/OxDc) [46].

GC–TCD can efficiently detect CO_2_ formed in the headspace of an EBFC, providing more accurate results regarding how much fuel has been oxidized [36,55]. TCD is used to detect volatile compounds that show low response in other detectors, including UV detectors and RID. Compounds with good thermal conductivity, such as ammonia, hydrazine, and CO_2_, are the most suitable for this analytical technique and may also be applied for quantification [56,57]. Headspace GC–TCD has been demonstrated to detect CO_2_ efficiently during complete glycerol oxidation [34,36,55]. Our research group has applied GC–TCD to identify CO_2_ in a BFC containing a hybrid system based on MWCNT-COOH/pyrene-TEMPO/OxDc, which confirmed that this bi-catalytic system collected 12 electrons from ethanol by completely oxidizing it to CO_2_ [34].

Apart from detecting CO_2_ and confirming complete fuel oxidation in either solution or the headspace, HPLC [32,33], NMR [46], and GC-TCD [34] are essential to understand how the catalyst bioelectrode interacts or reacts. These analytical techniques allow for the concentration of the by-products formed during fuel oxidation to be calculated and for the mass balance and faradaic efficiency to be determined. This is vital when it comes to understanding the mechanism and the contribution of the individual catalysts to the whole system. Straightforward use of analytical techniques for long-term electrochemical applications will be routinely required in any EBFC laboratory.

## 6. Applications of Ethanol EBFCs for Biosensing

### 6.1. Enzymatic Biosensors

Compared to chemical catalysts, enzymes have a high level of specificity and selectivity for the substrate. For this reason, modifying a bare surface with enzymes has become one of the most active areas of electroanalysis [58]. The relatively low stability of enzymes for use as biosensors can be easily overcome by choosing the appropriate conditions of pH and temperature and by immobilizing the enzyme properly [59]. Covering all of the literature concerning biosensors is out of the scope of this review, but several reviews have detailed the use of one or more enzymes as a biological component of the biosensor. Numerous studies have reported the application of ethanol biosensors [60,61,62,63]. In these investigations, two main enzymes, namely alcohol oxidase (AOx) [60,61] and ADH [62,63], have been successfully employed to determine alcohols. Chui et al. developed an amperometric ethanol biosensor by immobilizing ADH on the surface of a poly(vinyl alcohol)–multi-walled carbon nanotube (PVA–MWCNT) composite. The ethanol biosensor showed high sensitivity (196 nA mM^−1^) and fast response (about eight seconds) to detect ethanol, which enabled its use in real samples such as beer, red wine, and brandy [62]. A screen-printed carbon electrode modified with 5% cobalt phthalocyanine (CoPC-SPCE) and containing AOx was applied as an ethanol amperometric biosensor [60]. The amperometric technique showed good performance and provided high precision and reliability for ethanol detection in beer [60]. For ethanol detection, a search of the Web of Science database using the keywords sensors, enzyme, and ethanol retrieved more than 10,886 entries from the Web of Science core collection. Figure 4A depicts the timeline of these publications and shows that interest in this field has increased markedly since the 1990s. Figure 4B shows that the subject was mainly investigated in Japan, the USA, China, and Germany.

#### 6.1.1. Electrochemical Biosensors

Electrochemical biosensors are characterized by their simplicity, sensitivity, reliability, and fast response. These biosensors provide exceptionally low detection limits and operate in a wide concentration range. Due to all these advantages, electrochemical biosensors constitute most of the developed biosensors [64].

In recent years, research into EBFCs has focused on developing enhanced hybrid architectures that can completely oxidize the fuel and can collect the maximum energy/electrons per fuel molecule. Obtaining EBFCs with high energy efficiency will allow these promising systems to be applied in the production of EBFC biosensors through energy management [30,65,66].

#### 6.1.2. Ethanol Self-Powered Biosensors (ESPBs)

On the basis of some literature reviews, ethanol self-powered biosensors (ESPBs) have increasingly attracted researchers’ interest due to their practical applications related to health, food analysis, and environmental monitoring [67,68,69]. Most reviews on biosensors have reported that these systems show high sensitivity and selectivity and provide fast response [31,68,69]. However, some issues must be overcome. In some cases, the low stability of these systems with respect to specific targets was reported as a recurring problem [66,68]. Therefore, new fuel cell designs that include structured materials and promising organic/biological catalysts must be employed to improve the stability of enzymatic self-powered biosensors [67,69]. The first self-powered enzymatic biosensor was based on the consumption of glucose; it involved an oxidoreductase enzyme at the anode and cytochrome c oxidase at the cathode (to reduce oxygen) [58]. Devices to detect other substrates such as glucose [58,70], lactate [58,71], cholesterol [72,73], and drugs and antibiotics [74] were recently developed. As stated previously, to a lesser extent, ethanol was used as an analyte for application in biosensors [68,75,76,77]. Schuhmann et al. [31] reported on a self-powered biosensor device based on ethanol/O_2_ biofuel cells consisting of a bioanode modified with a β-nicotinamide adenine dinucleotide (NAD^+^)-ADH/redox polymer and a biocathode modified with AOx and horseradish peroxidase (HRP) to detect ethanol in a liquor. To improve the bioanode performance, the authors used a phenothiazine dye-modified redox polymer to recycle and reduce the NAD^+^ cofactor overpotential. The chronoamperometric experiments revealed a linear current response for ethanol concentrations ranging from 0.1 to 1.0 mM. The proposed ethanol biofuel cell exhibited a high open-circuit voltage (OCV) of approximately 660 mV, arousing interest in the development of new ethanol self-powered energy conversion technology [31].

Gao et al. proposed another self-powered ethanol biosensor [78]. To construct the bioelectrode, the authors used liquid-crystalline lipidic cubic phases (LCPs) composed of monoolein (MO) as a hosting matrix to co-entrap ADH and the electrocatalyst toluidine blue (TB). The ethanol biosensor had a detection limit of 0.09 mM and linear ethanol concentration range up to 15.6 mM. The authors employed the system to detect ethanol in human serum with good reproducibility. An investigation into the performance of the ethanol/air EBFC by power density tests provided an OCV and maximum power density of 0.53 V and 12.0 μWcm^−2^, respectively [78].

Our group employed the hybrid bi-catalytic architecture consisting of MWCNT-COOH/pyrene-TEMPO/OxDc [34] to develop a self-powered ethanol biosensor. Figure 5 illustrates the calibration curve of this hybrid system at various ethanol concentrations derived from chronoamperometric experiments.

The sensor response to a large ethanol concentration range (0–2500 mM) showed that the current density (*j*) increased after successive additions of different ethanol concentrations. After 2000 mM ethanol, the system became saturated and *j*_max_ remained constant. From 0 to 100 mM ethanol, the biosensor presented good linearity with a linear relationship (R^2^ = 0.9906) between the ethanol concentration and the current. The ethanol detection limit was 0.10 mM. These values were comparable to data achieved with other applied methods for self-powered ethanol biosensors [31,78].

The bioelectrode ability to generate high current densities in a wide range of ethanol concentrations allowed us to obtain a self-powered ethanol biosensor [30,31,75,79]. Figure 6 shows the power curves of the MWCNT-COOH/pyrene-TEMPO/OxDc,ethanol//Pt/C,O_2_ biofuel cell at different ethanol concentrations. The power density (Figure 6A) and the maximum current density, I_max_, (Figure 6B) increased linearly with the ethanol concentration from 0 to 100 mM ethanol. In the presence of 5 mM ethanol, the power density and current density of the self-powered sensor reached values as high as 80 μW cm^−2^ and 310 μA cm^−2^, respectively. The results showed that the current density and power density were a function of ethanol concentration, which clearly demonstrated that the BFC acted as a self-powered biosensor.

Table 1 summarizes the power curve results obtained for different ethanol biofuel cells. The results indicate that the designed MWCNT-COOH/pyrene-TEMPO/OxDc system achieved higher power density values compared to other systems evaluated for ethanol biofuel cells (BFCs). It is noteworthy that the hybrid catalytic architecture can be potentially employed in small bio-powered devices, which opens up opportunities for various biosensing applications.

### 6.2. Approaches to Improve the ESPB Technology: Supercapacitor/Biofuel Cell Hybrid Device

Many research groups have been encouraged to make BFCs a more accessible technology; these cells are directly associated with the development of ESPB devices, which provide ways to obtain clean and renewable energy and have potential use as an alternative energy source for low power electronic devices [65,80,81].

Despite recent advances, employing BFCs in electronic devices is no easy task: BFCs have limitations, such as insufficient stability, power, and energy production to promote autonomous energy sources. Although an increasing number of publications have reported satisfactory power densities, the values obtained to date are far from practical use in long-term applications.

A possible solution to develop efficient ESPBs is to develop devices that can generate/store energy when they are coupled with supercapacitors (SC) [82]. SCs are high-power electrochemical energy storage systems with high capacitance electrodes that can be charged and discharged by fast and reversible processes, thereby allowing an almost unlimited number of charge/discharge cycles [83,84]. SCs can function as a battery (high energy storage capacity) while providing capacitor performance (fast charge and power supply) [85,86].

Therefore, in SC-EBFCs, the electrode internal capacitance is used to accumulate the electrical charge generated at the biobattery (organic catalyst and enzyme) [87]. Some examples of hybrid biodevices that integrate supercapacitors such as biobatteries and biosensors and use enzymatic systems have been developed recently [88,89,90]. Pankratov et al. [91] developed a glucose self-charged biocapacitor based on graphite foil modified with a polyaniline/carbon nanotubes (CNT’s) composite as the capacitor; the EBFC consisted of nanobiostructures based on three-dimensional gold nanoparticles (AuNPs) at the anode and a AuNPs biocathode. The authors achieved a maximum power density of 1.2 mW cm^−2^ at 0.38 V. This conjugated system proved an efficient glucose self-charging device that generated 170 times higher power density compared to the EBFC alone [91].

The supercapacitive properties of CNTs used to store charges in SC-EBFCs have been widely reported [80,92]. Agnes et al. [80] reported on a hybrid SC-EBFC based on a matrix of compressed CNT pellets as SC combined with GOx at the anode. The system displayed high-pulsed power discharges. In addition, the generated energy was stored within the CNT matrix, which enhanced the stability of the system. The hybrid system produced 40,000 pulses for five days, providing 2mW per pulse of discharge [80].

Considering the excellent results obtained with the hybrid systems and the advantages of employing ethanol, building a hybrid SC-EBFC system would be interesting because it would allow environmentally sustainable SC-EBFCs to be developed for applications that require different power/current density ranges and durations of operation, enabling portable devices, such as biosensors, to be produced [65,81,82,93].

Attempts to employ the hybrid system MWCNT-COOH/pyrene-TEMPO/OxDc to prepare a self-powered biosensor device to detect ethanol will be reported in the future. However, Figure 7 shows the proposed representative model for a MWCNTCOOH/pyrene-TEMPO/OxDc,ethanol//Pt/C,O_2_ device for electric power generation based on ethanol oxidation. The energy generated in the EBFC (a) is charged into the capacitor via a charge pump-integrated circuit (b) until the capacitor (c) reaches maximum capacity. Thus, the rate at which the capacitor is charged may be directly proportional to the efficiency of the bioelectrocatalytic ethanol oxidation reaction. When the capacitor voltage reaches a set value, the stored electrical energy can be discharged from the capacitor to activate a device such as an LED bulb.

## 7. Conclusions

Enzymatic biofuel cells have advanced in terms of catalytic activity and energy production rate, but several approaches have been proposed to overcome the problems related to EBFC performance and stability. Remarkably, hybrid bi-catalytic bioelectrodes containing an organic catalyst (pyrene-TEMPO), a decarboxylase enzyme (OxDc), and modified carbon nanotubes have been able to increase the bioelectrode surface area, thereby enhancing the energetic performance of the hybrid system and improving the EBFC lifetime.

We have shown that the biobattery composed of MWCNT-COOH/pyrene-TEMPO/OxDc,ethanol//Pt/C,O_2_ has potential use in small bio-powered devices with linear response ranges toward ethanol (0.1 mM). This is an improvement that brings EBFCs closer to biosensing applications in the real world. A further improvement could be the use of hybrid SC-EBFC systems to overcome the issues of EBFCs. Such hybrid systems could be an alternative to achieving high-performance hybrid EBFC-based self-powered biosensors without the need for an external electrical power supply. This could allow for the development of ethanol EBFC biosensors that may be valuable for the determination of ethanol in real samples.

Future research on ethanol EBFCs for biosensing should focus on engineering approaches capable of improving the capacitance of the hybrid system and therefore increasing the energy storage and efficiency of the EBFC, which will enhance the analytical performance of the biosensor.

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
