# Peer review of "Ethanol Biofuel Cells: Hybrid Catalytic Cascades as a Tool for Biosensor Devices"

_biosensors, 2021, doi:10.3390/bios11020041_

Round 1

Reviewer 1 Report

This is an excellent review of enzymatic biofuel cells. It is well written and requires only the following minor changes.

  1. Line 44, the references “[3][11][12]” should be cited as “[3,11,12]”. This applies also to references cited in lines 64, 67, 70, 83, 84, 86, 88, 91, 93, 95, 100, 114, 134, 136, 140, 160, 206, 231, 234, 237, 241, 242, 247, 264, 265, 294, 298, 299, 301, 304, 307, 310, 350, 352, 384, 394, 396, 400, 408, 417
  2. Line 66, “nontoxic” should be “non-toxic”.
  3. Line 156, heading should be “Analytical Techniques Employed with Ethanol EBFCs”.

The manuscript can be accepted for publication after these minor revisions have been made.

Reviewer 2 Report

In this review, the authors presented the recent technical achievements on enzymatic biofuel cells and their applications as biosensors. This manuscript is well written and covered most recent literatures in this field. Publication is recommended after addressing the following points:

  1. Remove “1. Introduction” and change “1.1 enzymatic biofuel cells” to “1. enzymatic biofuel cells”, since there is 1.2 session.
  2. Similarly, remove the line 118-119, since there is no 3.2 session.
  3. Line 123: abbreviation LPEI is first mentioned here without its whole form, though the whole form is listed in Table 1 later.
  4. Line 124: “This system cleaved the ethanol C-C bond”. Actually, this system cleaved the acetic acid C-C bond.
  5. Line 220: “nuclear magnetic resonance” is unnecessarily underscored.
  6. Please reformat Table 1. Journal names are shown in different colors, with some of them underscored. Using the abbreviations of Journal names is recommended.
  7. Line 348: (R2 9906) --> : (R2 = 0.9906)
  8. Figure 7: The bilirubin oxidase on the cathode is only mentioned in line 33 without the abbreviation Clarifying the names of the two enzymes immobilized on the anode and cathode in the figure caption is appreciated.
